biophysics

DNA, electron transfer, nonlinear models, adiabatic polaron, solitons, DNA topology

**Author for correspondence:**
Slobodan Zdravković
e-mail: szdjidji@vin.bg.ac.rs

# A review on nonlinear DNA physics

Dalibor Chevizovich[1], Davide Michieletto[2], Alain Mvogo[3], Farit Zakiryanov[4] and Slobodan Zdravković[1]

[1]Institut za nuklearne nauke Vinča, Univerzitet u Beogradu, 11001 Beograd, Serbia
[2]School of Physics and Astronomy, University of Edinburgh, Peter Guthrie Tait Road, Edinburgh EH9 3FD, UK
[3]Laboratory of Biophysics, Department of Physics, Faculty of Science, University of Yaounde I, PO Box 812, Cameroon
[4]Bashkir State University, 32 Zali Validi Street, 450076 Ufa, Republic of Bashkortostan, Russia

SZ, 0000-0002-5893-6819

The study and the investigation of structural and dynamical properties of complex systems have attracted considerable interest among scientists in general and physicists and biologists in particular. The present review paper represents a broad overview of the research performed over the nonlinear dynamics of DNA, devoted to some different aspects of DNA physics and including analytical, quantum and computational tools to understand nonlinear DNA physics. We review in detail the semi-discrete approximation within helicoidal Peyrard–Bishop model and show that localized modulated solitary waves, usually called breathers, can emerge and move along the DNA. Since living processes occur at submolecular level, we then discuss a quantum treatment to address the problem of how charge and energy are transported on DNA and how they may play an important role for the functioning of living cells. While this problem has attracted the attention of researchers for a long time, it is still poorly understood how charge and energy transport can occur at distances comparable to the size of macromolecules. Here, we review a theory based on the mechanism of 'self-trapping' of electrons due to their interaction with mechanical (thermal) oscillation of the DNA structure. We also describe recent computational models that have been developed to capture nonlinear mechanics of DNA *in vitro* and *in vivo*, possibly under topological constraints. Finally, we provide some conjectures on potential future directions for this field.

## 1. Introduction

The deoxyribonucleic acid (DNA) is among the most interesting and intriguing biomolecules in nature. The interest in its structure and dynamics is primarily due to the important role

that this molecule plays in life. Each molecule of DNA is a double helix (duplex) formed by two complementary chains of nucleotides [1]. Association of single chains in the duplex is due to the specific hydrogen bonding of adenine with thymine and guanine with cytosine.

In the last few decades, molecular biology has made significant progress in studying the structure and dynamics of biological molecules including DNA. In order to better quantitatively understand these new results as well as to design better experiments, models that have solid foundations on physical laws are strongly needed. At the same time, the development of nonlinear theories (which introduced new concepts such as self-organization and solitons) provides a starting point for the analysis of complex systems which are characterized by the interactions of many components. For this reason, the application of nonlinear theories to understand the physics and biology of DNA is a promising field of research. It should be also mentioned that most of the biological processes in which DNA is involved, occur at length scales that bridge the submolecular (approx. 1 nm) and microscopic (10 µm) scales. Therefore, in order to achieve a full understanding of processes involving the DNA, this must include models based on both quantum and classical principles.

For most models explained in this review, an elementary unit is a nucleotide, which means that its atomistic structure is neglected. In fact, some analytical treatments often focus on simple models because they make more transparent the physics and simplify its mathematics, but, of course, there are also models that go beyond the level of nucleotides and can capture the full atomistic structure of DNA (some of these models will be discussed later on).

Constituent parts of each nucleotide are sugar, phosphate and base. Interactions between two neighbouring nucleotides belonging to the same strand are strong covalent bonds, and they are modelled by harmonic potentials. On the other hand, interactions between the nucleotides belonging to the different strands are weak hydrogen bonds (H-bonds). The existence of a strong force means that displacements along the direction of this force are very small. Consequently, we can assume that attractive and repulsive forces are almost equal and the corresponding potential energy, or potential for short, should be modelled by a symmetric function. A typical example is the well-known function $f(x) = kx^2/2$. Such potential is called harmonic. Its first derivative represents a force, which is obviously a linear function. On the other hand, if these forces are weak, the corresponding displacements, i.e. the values of $x$, are big and the repulsive and attractive forces are not equal anymore. To model such potentials, we need non-symmetric functions. A common example is the function $F(x) = D\,(e^{-a\,x} - 1)^2$, called Morse potential. It will be explained in more detail in §3. The first derivatives for negative and positive $x$, representing the repulsive and attractive forces, are not equal. For very big distance between the interacting particles, the first derivative is zero, which means that the particles do not interact anymore. This potential is not harmonic and the appropriate force is not a linear function. The models including at least one anharmonic interaction are called nonlinear. Therefore, the weak interactions are the sources for the nonlinear terms and, consequently, such systems are nonlinear. As these weak forces are common for biological systems, we focus on the nonlinear models only.

In addition to H-bonds, the stability of the duplex is supported by another type of non-covalent attraction, the so-called stacking. This kind of weak interaction occurs between neighbouring bases of the same chain and is of crucial importance for DNA twisting [2,3]. Namely, Watson & Crick showed that both hydrogen bonding between A–T and G–C base pairs and stacking interactions between adjacent bases result in helical duplexes when two sequences are complementary [3].

At this stage, it is important to highlight that the double-helical structure of DNA is far from static, indeed it can 'open' or 'denature', when one or more neighbouring nucleotide pairs break the complementary H-bonds that keep them in the duplex. Opening of DNA chain provides access of enzymes to the bases; therefore, open states actively participate in specific DNA–protein interactions.

In what follows, we briefly review some theoretical approaches to study the nonlinear dynamics of DNA (§2) and we explain, in some more detail, the helicoidal Peyrard–Bishop (HPB) model (§3). We show that this model can explain some important features of DNA, e.g. local opening and DNA transcription. As a mathematical tool, a semi-discrete approximation is demonstrated. This procedure yields to so-called breather solutions, which are local denaturation bubbles which move along the chain, or localized modulated solitary waves.

In §4, we focus on a process that occurs on smaller scales. Specifically, we review the efforts made to explain the high efficiency of electronic transport at distances comparable to the length of a DNA molecule. Several different models have been proposed over the past few decades and most of them are based on the assumption that charges can be self-trapped due to the interaction with mechanical

oscillations of the macromolecular chain and forms a polaron state. As we shall see, since this process takes place at the submolecular level, its description requires a quantum mechanical approach. In this section, we will see how this process is connected to the phenomena of stable electron migration along double-chain macromolecular structures like DNA. It should be highlighted that this is a peculiar aspect of DNA, i.e. that it is involved in important biological processes that span a wide range of length scales, from submolecular to microscopic ones.

In §5, we describe some computational approaches to understand the physics of double-stranded DNA. As we discuss later in this paper, this is a challenging task: currently, only very short DNA molecules can be simulated at full atomistic detail and some coarse-graining is necessary in order to reach realistic DNA lengths. The advantage of computational models is that they can deal with full nonlinear interactions and complex situations such as topologically constrained DNA molecules, i.e. closed into a loop or tied into a knot, which are relevant *in vivo*.

Finally, we review some experimental results and present some potential ideas for micro-manipulating experiments on single DNA molecules informed by the theoretical results recapitulated in this review. We also highlight that this is an unusually broad review; the problem of understanding phenomena associated with nonlinear DNA mechanics can be tackled from many perspectives and we thus feel that an unfamiliar reader entering this field will benefit from the broad context provided in this review.

# 2. Theoretical studies of DNA open states

The construction and study of nonlinear models has a long tradition in the life sciences. Suffice it to mention the Volterra model (predator–prey) and the Belousov–Zhabotinsky equations. To date, dozens of different mechanical models and their versions have been developed to describe nonlinear DNA dynamics at various spatial and temporal scales. These are continuous or discrete models, models that take into account the helical structure or neglect it, account for the movements of every atom or simulate the motion of only the main subunits, homogeneous models or models that describe a specific sequence of bases. Very comprehensive surveys can be found in [4–6]. The simplest structural model of a fragment of DNA is the model of an elastic rod [5,7,8]. More advanced models are helical double rod-like models [5]. In both cases, the rods can be either uniform or discrete. According to these simple models, plain waves propagate along the chain.

A new direction in DNA research began when Englander *et al*. [9] suggested that nonlinear effects might play an important role in the DNA dynamics. Indeed, nonlinear effects may focus the vibration energy of DNA into localized soliton-like excitations. According to the model, DNA can be represented as two linear chains of pendulums (the bases) connected by springs to the sugar-phosphate backbones, as shown in figure 1. The total energy in terms of the rotation angles $\theta_n$ is given by the following Hamiltonian:

$$H = \sum \left[ \frac{mh^2}{2}\left(\frac{\mathrm{d}\theta_n}{\mathrm{d}t}\right)^2 + \frac{S}{2}(\theta_n - \theta_{n-1})^2 + mgh(1 - \cos\theta_n) \right], \tag{2.1}$$

where $m$ and $h$ are mass and length of the pendulum, respectively [9]. All sums in this paper run over all the nucleotides. The nonlinearity comes from the cosine function. The Hamiltonian (2.1) brings about a differential equation

$$mh^2 \frac{\mathrm{d}^2\theta_n}{\mathrm{d}t^2} = S(\theta_{n+1} + \theta_{n-1} - 2\theta_n) + mgh\sin\theta_n. \tag{2.2}$$

In the static limit, when its left-hand side is zero, a soliton solution of equation (2.2) is [9]

$$\theta_n = 4\arctan\left[\exp\left(\frac{2nl}{L}\right)\right], \quad L = 2l\sqrt{\frac{S}{mgh}}, \tag{2.3}$$

where $l$ is the distance between two neighbouring pendulums. The function $\theta_n \equiv \theta(n)$ is a kink soliton.

This procedure can be extended as the DNA molecule can be seen as a series of coupled double pendulums [10,11]. The first pendulums model the oscillations of the phosphate-sugar parts of nucleotides, while the second ones describe the oscillations of bases. It is interesting that these two angles, the one in equation (2.3) and the one corresponding to the second pendulum, represent

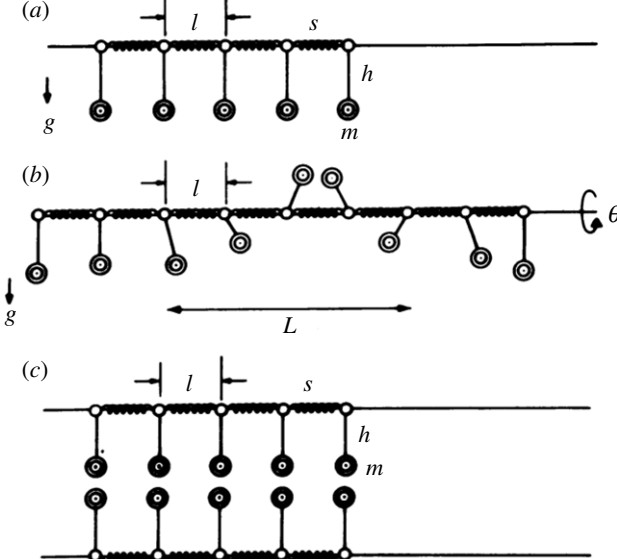

**Figure 1.** Mechanical analogue of a double helix, representing linear chains of pendulums (the bases) connected by springs (the sugar-phosphate backbones). (*a*) A single strand. (*b*) Soliton excitation mode, spreading of the excitation to a group of *L*. (*c*) The ground state of the double helix. (Reproduced by kind permission of Dr Englander from [9].)

topological and non-topological degrees of freedom, respectively [11]. This approach can be further extended describing inhomogeneous DNA chains [12].

The last three decades, beginning with the works of Davydov [13], are characterized by a particularly intensive use of methods of nonlinear physics to describe the various properties of biopolymers, including the use of nonlinear solitary waves to describe the dynamics of DNA.

A key problem in each model is a choice of degrees of freedom. Namely, the formation of the open states can be connected with either angular or translational displacements of the bases from their equilibrium positions. Accordingly, if we assume only one degree of freedom per nucleotide, we can choose either angular or translational variable as coordinate. Let us call appropriate models angular (torsional) and translational models, respectively. Of course, some extensions, i.e. models combining both kinds of coordinates, are possible [14–19]. The model mentioned above [9] is the angular one as well as the Y-model, introduced by Yakushevich [20] and has been further developed [21–26]. Like in the Englander's model, DNA dynamics is represented by kink solitons. Similar models have been used to study the dynamics of DNA under the influence of external forces [26–29].

This kind of solitary waves can also be obtained relying on the plane-base rotator (PBR) model. The model assumes that the sugar-phosphate parts of the nucleotides do not move, while the bases oscillate in the plane perpendicular to the helical axis around the backbone structure and the appropriate coordinates are angles $\Phi_n$ [30–32]. The model was proposed by Yomosa [33,34] and improved by Homma & Takeno [35]. The same procedure can be used to study completely inhomogeneous DNA chain [36]. Further improvement was done introducing an analogy between the PBR model and the Heisenberg's spin model for ferromagnets, which brings about the Hamiltonian [30,37]

$$H = \sum \left[ \frac{I}{2}\dot{\Phi}_n^2 + \frac{I'}{2}\dot{\Phi}'^2_n - Jf_n \cos(\Phi_{n+1} - \Phi_n) - J' f'_n \cos(\Phi'_{n+1} - \Phi'_n) + \eta \cos(\Phi_n - \Phi'_n) \right], \qquad (2.4)$$

where $I$ and $I'$ are the momenta of inertia. The prime notation has been introduced to distinguish the two strands. The meanings of all the parameters in equation (2.4) can be found in aforementioned references. Using some approximations, we reach the following kink and antikink solutions

$$\Phi = 4 \arctan \exp[\pm \kappa(z - vt)], \qquad (2.5)$$

where the parameter $\kappa$ depends on the soliton speed [30,37]. It might be interesting to compare the soliton velocities in single- and double-stranded DNA. If the velocity of double-stranded DNA is larger than the one of single-stranded chain then the two strands are coupled antiferromagnetically. Otherwise, the two DNA strands are coupled ferromagnetically [37]. It is possible to incorporate viscosity into the model [38]. Also, the model can be used to study protein–DNA interaction [38].

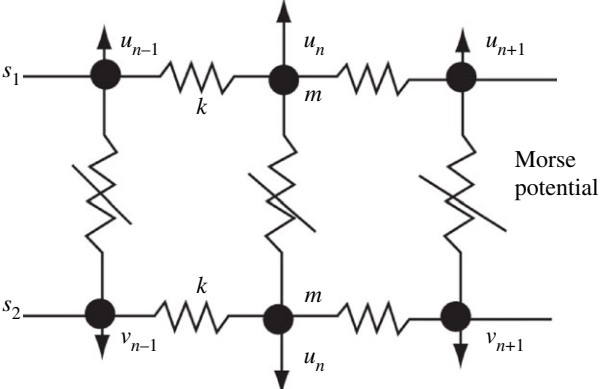

**Figure 2.** A segment of DNA molecule showing both covalent interactions, determined by the parameter $k$, and hydrogen bonds, modelled by Morse potential.

# 3. Helicoidal Peyrard–Bishop model for nonlinear DNA dynamics

The models explained above are angular ones. They predict that kink solitons move along the DNA chain. A well-known example of the translational models is the Peyrard–Bishop (PB) model [39,40], as well as its two extended versions. They are a helicoidal PB (HPB) and Peyrard–Bishop–Dauxois (PBD) models, explained below. The main requirement for the translational approach is the description of complementary H-bonds through a nonlinear potential. This condition is essential for the localization of energy. It was stated in §1 that the longitudinal interactions are strong covalent bonds modelled by harmonic potentials. The transverse interactions between the nucleotides belonging to the different strands are weak hydrogen interactions, requiring an anharmonic potential. An example, used for the PB model, is the Morse potential

$$V_M(u_n - v_n) = D\left[e^{-a(u_n - v_n)} - 1\right]^2, \tag{3.1}$$

where the parameters $D$ and $a$ are the depth and inverse width of the Morse potential well, respectively. The coordinates $u_n$ and $v_n$ are transverse displacements of the nucleotides at the position $n$ from their equilibrium positions along the direction of the hydrogen bond. In the nearest neighbour approximation, the Hamiltonian for DNA is

$$H = \sum \left\{ \frac{m}{2}(\dot{u}_n^2 + \dot{v}_n^2) + \frac{k}{2}[(u_n - u_{n-1})^2 + (v_n - v_{n-1})^2] \right\} + V_M(u_n - v_n), \tag{3.2}$$

where $m = 300$ amu $= 5.1 \times 10^{-25}$ kg is the nucleotide mass, $k$ is a constant of the harmonic interaction, while $\dot{u}_n$ and $\dot{v}_n$ represent the appropriate velocities. The first two terms represent the kinetic and potential energies of the longitudinal spring, respectively. In this case, nonlinearity is coming from the Morse potential, describing the weak hydrogen interaction. It is convenient to introduce new coordinates representing the in-phase and the out-of-phase transversal motions as

$$x_n = \frac{(u_n + v_n)}{\sqrt{2}}, \quad y_n = \frac{(u_n - v_n)}{\sqrt{2}}, \tag{3.3}$$

which transforms Hamiltonian (3.2) into

$$H = \sum \left\{ \frac{m}{2}(\dot{x}_n^2 + \dot{y}_n^2) + \frac{k}{2}[(x_n - x_{n-1})^2 + (y_n - y_{n-1})^2] \right\} + V_M(y_n). \tag{3.4}$$

According to equation (3.3) and figure 2, we can conclude that $x_n(t)$ and $y_n(t)$ describe the in-phase and out-of-phase oscillations of the nucleotide pair at the position $n$, respectively. In other words, $x_n(t)$ represents the oscillation of the centre of mass of the nucleotides, while $y_n(t)$ is their stretching. We do not explain the model in detail as this will be done through one of its improved versions. In fact, the PB model is a special case of the HPB model, which will be explained in what follows. It suffices now to point out that the function $x_n(t)$ represents a linear wave, while $y_n(t)$ is a nonlinear one. It is important to know that $y_n(t)$ is not temperature-dependent, but its mean value $\langle y \rangle$ is [39–41]. This model can explain denaturation of DNA, meaning that the quantity that can

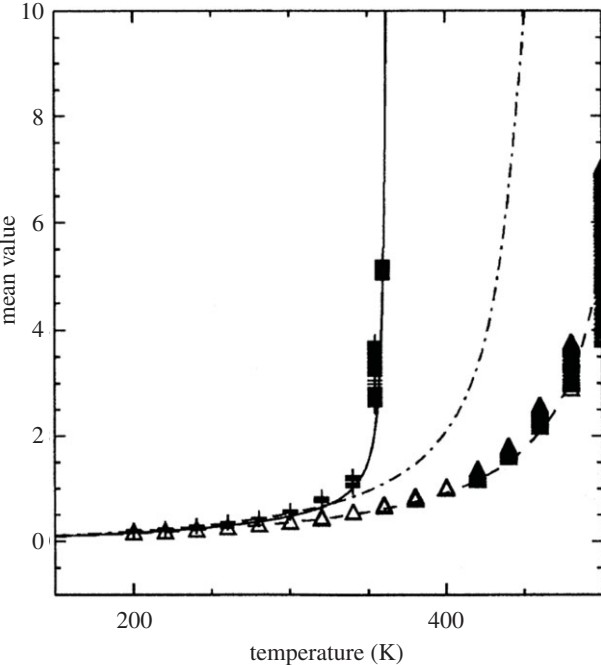

**Figure 3.** Variation of the mean value $\langle y \rangle$, i.e. the mean stretching of the hydrogen bonds, versus temperature for: $k = 0.04$ eV $\text{Å}^{-2}$, $a = 4.45$ $\text{Å}^{-1}$, $D = 0.04$ eV. The solid line corresponds to the anharmonic stacking interaction ($\alpha = 0.35$ $\text{Å}^{-1}$, $\rho = 0.5$). The dashed and dash-dotted lines correspond to two cases ($\rho = 0.5$ and $\rho = 0$) of harmonic stacking interactions ($\alpha = 0$), respectively. (Reproduced with permission from [44].)

capture the extent of the denaturation of DNA molecules in solution is the mean stretching of the hydrogen bonds [41]. The melting temperature, given by the numerical treatment, is significantly higher than the one corresponding to continuum approximation. This certainly indicates the large role of discreteness in DNA dynamics [41].

Before we proceed, we want to mention an interesting attempt to improve the modelling of the base-pair interaction. Namely, the term proportional to $-D[\tanh((u_n - v_n)/\lambda_s) - 1]$ can be added to the one given by equation (3.1) [42]. This is a solvent interaction potential, simulating the formation of hydrogen bonds with the solvent once the base-pair hydrogen bonds are stretched by more than a value $\lambda_s$ from their equilibrium values [42]. Also, different Morse potentials can be used instead of the one in equation (3.1) [43], which we do not elaborate in this review.

Two basic improvements of the PB model have been done so far. In this paper, we call them the HPB and PBD models, although the terminology in references has not been unique so far. In the PBD model, the harmonic potential energy has been replaced by the anharmonic one through

$$\frac{k}{2}(y_n - y_{n-1})^2 \quad \Rightarrow \quad \frac{k}{2}[1 + \rho\,\mathrm{e}^{-\alpha(y_n + y_{n-1})}](y_n - y_{n-1})^2, \tag{3.5}$$

where $\rho$ and $\alpha$ are constants [42,44–49]. This expression can be viewed as a harmonic interaction with a variable coupling constant [50].

Figure 3 shows how the mean value $\langle y \rangle$, i.e. the mean stretching of the hydrogen bonds, depends on temperature. It demonstrates an advantage of the PBD model over those assuming harmonic approximation. Namely, $\langle y \rangle$ is slowly increasing function up to a certain temperature when it sharply increases. This increase represents denaturation. In the case of the PBD model, denaturation is rather sharper and occurs at lower temperatures. To be more precise, the authors compared the two cases within the potential given by equation (3.5), that are $\alpha = 0$ (PB model) and $\alpha \neq 0$ (PBD model), and demonstrated advantage of the latter one. It would be interesting to study temperature dependences of $\langle y \rangle$ relying on the HPB model, as well as other ones. We believe that this research should be performed in future. We want to point out that temperature dependence of $\langle y \rangle$ also depends on the remaining parameters existing in the model, like $D$ and $a$, describing the Morse potential. Some combinations give very sharp increase even for the harmonic case as well as too low melting temperature [45].

Before we proceed, we want to mention a Peyrard–Bishop–Holstein (PBH) model [51,52]. This is a hybrid of the PBD model and the Holstein one [53]. The PBH model has been introduced as an appropriate framework for the description of polaronic effects for charge migration in DNA, and it will be explained in some more detail in §4.

Chronologically, the first improvement to the PB model, crucial for the present section, was done by Dauxois *et al.* [54] and this is what we call the HPB model. A new term, describing helicoidal interactions, was introduced in equation (3.2) and the Hamiltonian becomes

$$H = \sum \left\{ \frac{m}{2}(\dot{u}_n^2 + \dot{v}_n^2) + \frac{k}{2}[(u_n - u_{n-1})^2 + (v_n - v_{n-1})^2] + \frac{K}{2}[(u_n - v_{n+h})^2 + (u_n - v_{n-h})^2] + D\left[e^{-a(u_n - v_n)} - 1\right]^2 \right\}, \quad (3.6)$$

where $K$ is the harmonic constant of the helicoidal spring. To understand the new helicoidal term, we need to imagine the twisted DNA molecule. This means that, after a turn of $\pi$, the nucleotide belonging to one strand at the position $n$ will be close to both $(n+h)$th and $(n-h)$th nucleotides of the other strand [54]. As the helix (B-DNA) has a helical pitch of about 10 bp per turn [55], one can assume $h = 5$.

The complete mathematical analysis including a lot of mathematical details can be found in [56]. We review it very briefly here. Equations (3.3) and (3.6) bring about two decoupled dynamical equations of motion, linear and nonlinear. We restrict our attention on the nonlinear one, which is

$$m\ddot{y}_n = k(y_{n+1} + y_{n-1} - 2y_n) - K(y_{n+h} + y_{n-h} + 2y_n) + 2\sqrt{2}aD\left(e^{-a\sqrt{2}y_n} - 1\right)e^{-a\sqrt{2}y_n}. \quad (3.7)$$

Of course, this is a discrete partial differential equation and our goal is to obtain its solution, i.e. the function $y_n(t)$. We assume that the oscillations of the nucleotides are large enough to be anharmonic but still small enough for the transformation

$$y_n = \varepsilon \, \Phi_n; \quad (\varepsilon \ll 1) \quad (3.8)$$

to be valid. To solve equation (3.7), we use a semi-discrete approximation, which means that we look for the wave solutions of the form

$$\Phi_n(t) = F_1(\xi)e^{i\theta_n} + \varepsilon \left[F_0(\xi) + F_2(\xi)e^{i2\theta_n}\right] + cc + O(\varepsilon^2) \quad (3.9)$$

and

$$\xi = (\varepsilon nl, \varepsilon t), \quad \theta_n = nql - \omega t, \quad (3.10)$$

where $l = 0.34$ nm is the distance between two neighbouring nucleotides in the same strand, $\omega$ is the optical frequency of the linear approximation, $q = 2\pi/\lambda$ is the wave number, cc represents complex conjugate terms and the function $F_0$ is real. This is a modulated wave where $F_1$ is a continuous function representing the envelope, while the carrier component $e^{i\theta_n}$ is discrete. A mathematical basis for equation (3.9) is a multiple-scale method or a derivative-expansion method [57,58].

A rather tedious mathematics [56] brings about a dispersion relation

$$\omega^2 = \left(\frac{4}{m}\right)\left[a^2 D + k\sin^2\left(\frac{ql}{2}\right) + K\cos^2\left(\frac{qhl}{2}\right)\right], \quad (3.11)$$

an expression for the group velocity $V_g = d\omega/dq$, and so on. Of special importance is the fact that the functions $F_0$ and $F_2$ can be expressed through $F_1$, while the latter one is a solution of the well-known solvable nonlinear Schrödinger equation (NLSE)

$$iF_{1\tau} + PF_{1SS} + Q\,|F_1|^2 F_1 = 0, \quad (3.12)$$

where $\tau$ and $S$ are new time and space coordinates [39,56]. Here, $P$ and $Q$ are the dispersion coefficient and the coefficient of nonlinearity, respectively [39,56]. For $PQ > 0$, the solution of equation (3.12) is [54,59,60]

$$F_1(S,\tau) = A_0 \, \text{sech}\left(\frac{S - u_e \tau}{L_e}\right) \quad \exp\frac{iu_e(S - u_c \tau)}{2P}, \quad (3.13)$$

which brings about the final solution [56]

$$y_n(t) = 2A\text{sech}\left(\frac{nl - V_e t}{L}\right)\left\{\cos(\Theta nl - \Omega t) + A\text{sech}\left(\frac{nl - V_e t}{L}\right)\left[\frac{\mu}{2} + \delta\cos(2(\Theta nl - \Omega t))\right]\right\}. \quad (3.14)$$

The expressions for $A \equiv \varepsilon A_0$, $L \equiv L_e/\varepsilon$, the envelope velocity $V_e$, wave number $\Theta$ and frequency $\Omega$ are given in [56,61].

To plot the function $y_n(t)$, the values of all the parameters should be known or, at least, estimated. There are two groups of them, mathematical ($\varepsilon$, $u_e$, $u_c$) and intrinsic parameters ($k$, $K$, $a$, $D$, $q$). It was suggested [62] that the wavelength covers an integer number of the periods $l$, i.e.

$$q = \frac{2\pi}{\lambda}, \quad \lambda = Nl, \quad N \text{ integer.} \tag{3.15}$$

Hence, one can assume $N$ as the internal parameter instead of $q$. Note that dependence on $\varepsilon$ can be eliminated [56]. This is something that one could expect as this is a working parameter only, which we use to distinguish the big and small terms in the series expansion (3.9).

The problem with the mathematical parameters has practically been solved using the idea of a coherent mode [56,63], assuming that the envelope and the carrier wave velocities are equal, i.e.

$$V_e = \frac{\Omega}{\Theta}. \tag{3.16}$$

This means that the function $y_n(t)$ is the same at any position $n$. In other words, the wave preserves its shape in time, indicating possible stability [61,63]. Note that equation (3.14) is one-phase function if equation (3.16) holds.

The intrinsic parameters have not been experimentally determined yet and we must be satisfied with their estimations only. Estimation of a value of a certain parameter is nothing but determination of its possible interval. Of course, increasing knowledge brings about narrower intervals. Finally, a certain combination should be picked, which allows us to plot the function $y_n(t)$, to calculate the solitonic speed $V_e$, its width $\Lambda$ that we define as $2\pi/\Lambda = 1/L$, etc. Note that different combinations may bring about more or less similar results.

The values of the parameters $a$ and $D$ were estimated according to both theory and experimental data in [63]. Also, we offered a proposal to experimentally determine them using single-molecule manipulation technique [64]. To the best of our knowledge, the most comprehensive and exhaustive analysis was performed in [65]. A possible set of the parameters could be [65]

$$a = 1.2\,\text{Å}^{-1}, \quad D = 0.07\,\text{eV}, \quad k = 12\,\text{N m}^{-1},$$
$$K = 0.08\,\text{N m}^{-1}, \quad N = 10, \quad \eta = 0.47. \tag{3.17}$$

Figure 4 shows the nucleotide pair stretching as a function of position, i.e. the function $y_n(t)$ given by equation (3.14). The set (3.17) was used as well as the values for $m$ and $l$ mentioned above. This is a localized modulated wave usually called breather. One can see that the breather covers about $\Lambda = 30$ nucleotide pairs. A discussion of this value can be found in [56].

The model explained above assumes homogeneous DNA chain, which might not be quite correct. For example, adenine and thymine are connected by double and guanine and cytosine by triple hydrogen bonds. Hence, one can expect that the corresponding Morse potential depths are $D$ and $1.5D$. A crucial question is whether the wave characteristics like amplitude, velocity, etc. drastically change whenever the wave reaches a new type of the nucleotide pair. If so, the soliton would not be stable. Fortunately, it was shown that DNA dynamics, i.e. the breathers' characteristics, are not sensitive to these inhomogeneities [66]. Namely, the breathers are not substantially affected by spatial inhomogeneities of the DNA sequence, but the kinks are [22].

Any model is good, more or less, if it can explain something or predict a possible experiment. The HPB model can explain local opening of DNA, the well-known fact which happens during transcription. From equation (3.7) and the appropriate linear one, we can easily obtain dispersion relations $\omega \equiv \omega_y = \omega_y(q)$, given by equation (3.11), and $\omega_x = \omega_x(q)$ [56]. These frequencies also depend on the parameters $k$, $K$, $a$ and $D$, and, in general, are not equal. They were compared [54] and it was speculated that their equality could represent a resonance mode [67]. The idea was further developed in [68,69]. Note that the relevant frequency is $\Omega$, existing in equation (3.14), rather than $\omega_y$ [68]. Here is a brief explanation of the resonance mode. The segments of DNA chain, where transcription occurs, are surrounded by RNA polymerase (RNAP) molecules. It was explained above that the mentioned parameters describe chemical bonds. This means that, due to the presence of RNAP, the interaction between the nucleotides belonging to the same pair is changed. In other words, the Morse potential at these segments is different from the potential at the rest of the molecule. The decrease of $a$ and $D$ brings about the values of the relevant frequencies being closer to each other. The resonance means

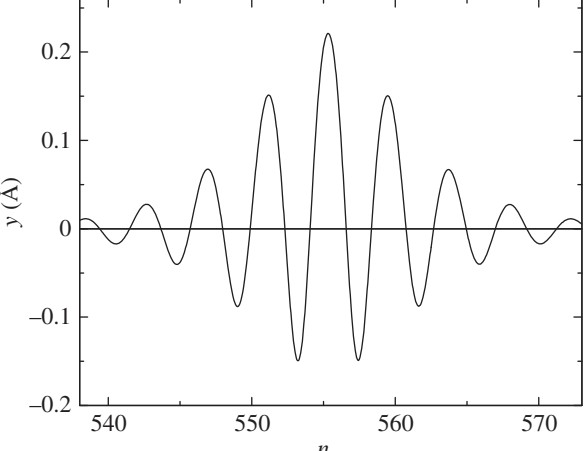

**Figure 4.** The nucleotide pair stretching at $t = 100$ ps for: $a = 1.2$ Å$^{-1}$, $D = 0.07$ eV, $k = 12$ N m$^{-1}$, $K = 0.08$ N m$^{-1}$, $N = 10$, $\eta = 0.47$.

that the positive values in figure 4 become very high while the negative disappear [68], which is nothing but local opening.

It may be interesting to point out that resonance is not possible if $K = 0$ [68]. This means that helicoidal structure provides resonance, which shows the advantage of the HPB model over the PB one.

Obviously, the infinitely large amplitude in the case of the resonance mode would represent destruction of the molecule. This should not bother us as the frictional forces have been neglected so far. To provide more realistic approach, we should take viscosity into consideration. This can be done by adding a viscous force

$$F_v = -\gamma \dot{y}_n, \tag{3.18}$$

to equation (3.7), where $\gamma$ represents a damping coefficient [70–72]. The optical frequency and the group velocity become [56,73]

$$\omega_\gamma + i\chi = \sqrt{\omega^2 - \chi^2}, \quad V_\gamma = \frac{\omega V_g}{\sqrt{\omega^2 - \chi^2}}, \quad \chi = \frac{\gamma}{2m}. \tag{3.19}$$

Following the procedure used when viscosity was neglected, we obtain equation (3.12) again. A crucial point is the fact that the nonlinear parameter $Q$ is now complex. This is why equation (3.12) cannot be solved analytically anymore. A numerical solution is still very interesting from biological point of view [73]. The obtained wave looks like the envelope of the soliton in figure 4 with smaller amplitude. This means that viscosity destroys modulation and the wave is bell-type soliton. This has very important biological implication [73], which we explain here briefly. Normally, we assume that the most important contribution to viscosity comes from the RNAP, surrounding the DNA chain at some segments that are being transcribed [74], as shown in figure 5. One can see that one of the two DNA strands serves as a template for synthesis of a new RNA strand. Two basic requirements should be satisfied for transcription to occur. First of all, the DNA nucleotides, belonging to the different strands, should be far away for some time. This is local opening and we explained that this is the resonance mode. During this period of time, the DNA and RNAP nucleotides interact. For successful transcription, this interaction should last long enough. This means that the breather, i.e. $y_n(t)$, at these segments, should have small enough frequency. The most appropriate would be zero frequency which is nothing but the modulated wave, which is the solution of equation (3.12) when viscosity is taken into consideration [73]. Therefore, it was found that viscosity causes demodulation, ensuring the long-lasting interaction between DNA and RNA nucleotides [73]. All this certainly shows that the HPB model is very important and convenient for description of DNA dynamics.

It might be interesting to point out one more contribution to studying DNA–RNA transcription. This is analytical research based on the idea of demodulated standing solitary waves existing in the relevant segments [75]. Namely, the idea was that the solitary wave, moving along the chain, transforms into a demodulated one at these segments. The second idea was that the wave becomes a standing one due

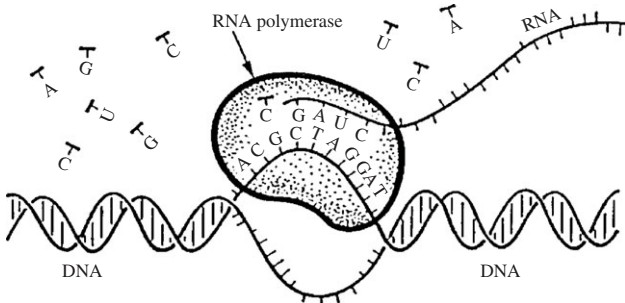

**Figure 5.** Transcription of DNA into RNA. Note the structural anisotropy of the helix, which features a 'major' and 'minor' groove. This will be recalled in §5.

to interaction with DNA surrounding. It was explained why this is biologically convenient and showed that the results match the experimental ones [75].

There have been a variety of attempts to improve the models mentioned above. One important example is introduction of different Morse potential [76]. It is interesting to mention a model representing a combination of the Englander's and the PB models, which brings about the kink soliton [77] and that the PB model can be used to study discrete breathers and multibreathers as well as their stability [78]. Finally, we want to point out very important and promising two-component model, using both angular and translational coordinates [15,79]. This helicoidal model can be seen as an angular–translational one.

All models explained above are mechanical ones. However, statistical physics and thermodynamics can also be a basis for DNA modelling, especially if we are interested in temperature dependence of DNA phenomena, such as melting. In this respect, we refer the interested reader to the review [6] and references therein.

One should also note that stacking interaction has been neglected so far. This means that the Hamiltonians mentioned above, e.g. equation (3.6), do not include any term describing it. Stacking is the interaction between the nucleotides belonging to the same strand. Of course, the same holds for the covalent bond, which is much stronger and one could say that the stacking can be neglected. However, even though the stacking is weaker than the covalent bond, we do not think that we should be satisfied with neglecting it, because the nature of these interactions is different. The HPB, or any other model discussed above, may be improved including one more term in the Hamiltonian and, since the stacking is weaker than the covalent bond, this term should be an anharmonic one. Therefore, the problem with the stacking is very interesting and should be a topic of future research.

In summary, in this section, we explained the HPB model and semi-discrete approximation, assuming the nucleotides as elementary particles. In other words, their internal structure was neglected. The nucleotides are relatively big particles and, consequently, quantum mechanical effects may be neglected. However, in the next section, we study electron transport along DNA. The appearance of the electrons certainly requires quantum mechanical approach, which is explained in what follows.

# 4. Long-distance electron transport along DNA macromolecular chains

At the heart of many processes that take place in living cells are the processes of energy and charge transport in biological macromolecules, such as protein macromolecular chains (MCs), RNA, DNA, etc. [80–83]. Therefore, without understanding these processes, we cannot even understand how a living cell functions. On the other hand, in the last few decades, the potential applications of biomolecules including DNA in the constructions of nanoscale biosensors, as well as enzymatic tools to detect the damage in the genome, have been intensively investigated. In contrast with the nonlinear problems presented in other parts of this work, the mentioned processes take place at a submolecular level, so quantum mechanical effects may play an important role. In turn, the models describing these processes ought to include principles of quantum mechanics.

As it was mentioned, the problem of stable transport of the energy and a charged particle along a spine of the MC at the distances comparable to its length (so-called long-distance charge transport in biological MCs) plays an important role in the understanding of the functioning of biological systems at the molecular level, and is still not fully understood. For example, the process of highly efficient long-range electron migration along biological macromolecular chains is important in redox reactions accompanying photosynthesis and cellular respiration. This transfer takes place at macroscopic distances along the so-called electron transport chain in Krebs cycles in membranes of chloroplasts, mitochondria or cells, and occurs at physiological temperatures. One of the earliest attempts to explain such problems on the basis of quantum mechanical principles was proposed in the mid-1970s by Davydov [80–83]. According to Davydov model, the excess charge can be captured by the protein molecules, and due to the interaction with mechanical oscillations of macromolecular chain, it forms self-trapped (ST) or polaron state [84–87]. As a matter of fact, due to the interaction with structural elements of MC, it induces a local distortion of the molecular crystal and forms a potential well. So, the created complex entity, i.e. the electron surrounded by local lattice distortion, may propagate in a soliton form along the chain with minimal energy losses preserving its shape and velocity for a long time. Davydov's ideas were motivated by the stability possessed by soliton solutions. It should be noted that Davydov's research was primarily devoted to the problem of stable energy transport in proteins with an α-helix secondary structure, but over the time, this model has also found application in modelling other phenomena, such is long-distance electron transport in biomolecules, especially proteins. Apart from Davydov solitons, different types of stabile states can occur in macromolecular structures, which depend on the geometry of the structure as well as its physical properties [85].

According to the general theory of self-trapping phenomena [84–87], the character of ST states of an excess exciton is determined by the mutual ratio of the values of three basic energy parameters of the crystal structure or MCs. The first two parameters are the exciton (electron, vibron, etc.) bandwidth $2|J|$ (here, $J$ is the longitudinal electron intersite transfer integral, which represents resonant energy of the electron transfer between the neighbouring molecules within the particular chain) and characteristic phonon energy $\hbar\omega_C$ (determined by the phonon cut-off frequency $\omega_C$), characterizing, respectively, the time-scale of motion in excitation and lattice subsystems. The third parameter is lattice deformation energy $E_b$, which measures the strength of excitation–phonon interaction [53,85–90]. We should pay attention to the two limiting cases: the large and small polaron limits [85–87]. The first one is reached when both the excitation bandwidth and lattice deformation energy highly exceed the characteristic phonon energy ($2J, E_b \gg \hbar\omega_C$), i.e. in the adiabatic strong coupling regime. In that case, phonons are slow with respect to the excitation dynamics and form (essentially classical) large radius quasi-static potential well by which that excitation is trapped. Davidov's solitons belong to this type of polaron. On the other hand, in the non-adiabatic limit (when the characteristic phonon energy is larger than the exciton bandwidth, i.e. for $\hbar\omega_C \gg 2J$), the quantum nature of the phonons plays a crucial role, and small polaron (SP) formation takes place. In that case, a quasi-particle becomes dressed by a virtual cloud of phonons which yields a lattice distortion essentially located on a single site and which instantaneously follows the excitation motion [53,85–90]. The quasi-particle dressed by the virtual phonon cloud forms a small polaron whose properties are well described by performing the so-called Lang–Firsov (LF) transformation [86,89]. Of course, transport properties of such quasi-particles must be quite different from those of bare electron [53,88,91,92]. An additional complication arises for structures where the energy parameters do not correspond to the mentioned boundary cases. For such systems, the aforementioned limits can then provide only an approximate picture of the process of the ST of excitons. For example, a proper theoretical description of the vibron ST in biomolecules requires an approach that goes beyond the conventional strong coupling SP theories [90,92].

Additionally, the properties and the stability of the soliton formed in macromolecular system depend on geometrical properties of the structure. It is well known that a stabile soliton formation can arise in pure one-dimensional (1D) structure (i.e. in the single chain macromolecule). But, in the quasi-1D structures (like a collection of bonded parallel macromolecular chains), the inter-chain interaction may have strong impact on the soliton features [85,87]. Let us emphasize here that the electron–phonon interaction in quasi-1D structures may cause additional anisotropy of the electron bands due to the exponential reduction of the electron transfer integrals, which in final instance may cause soliton stabilization [93].

Despite the great efforts that have been made in recent decades, we have not achieved a full understanding of the processes of MC charge transport. The reasons should be sought in the fact that

even the approximate values of the mentioned parameters are not known. As a matter of fact, they are not determined directly by measurement, but indirectly through a proposed theoretical model. Consequently, the corresponding estimation of their numerical values can vary from those that correspond to the conditions of weak interaction of excitons with the phonon subsystem, up to those that belong to strong coupling interaction limit [94–98]! In addition, MCs in living cells do not have the forms of pure 1D chains, but also they are neither classical two-dimensional (2D) nor three-dimensional (3D) structures. Instead, they are more similar to the system of coupled chains. For that reason, it is important to study the influence of the inter-chain interaction to the soliton properties, and this question requires additional research. All of mentioned have renewed interest for the investigations of the properties of nonlinear excitations such as various types of solitons, polarons or bipolarons that can be formed in biological macromolecular structures.

At present, it is generally accepted that the long-distance electron transport in single MC chains can be explained in the framework of the adiabatic large polaron (soliton) mechanism. Such an assumption is usually based on the observation that in most conjugated polymers, typical value of the intersite transfer integral has the value of approximately $J = 2.5$ eV and typical value of the electron–phonon coupling parameter $\chi$ lies around the value $\chi = 4.5$ eV $\text{Å}^{-1}$ [99]. In the same time, phonon energies range between 0.12 eV in polyacetilene and 0.2 eV in the double-strand polyacene molecule [98]. Similarly, in the case of the α-helix protein macromolecules, electron bandwidth is approximately a few eV, while the maximal energy of the acoustic phonons is approximately 17 meV [100,101]. As was mentioned above, the soliton formation can occur mainly due to the process of the self-trapping of an excess charge (photogenerated electron, or such charge that was created upon doping mechanism, for example), and the creation of polaron quasi-particles, which then can perform the role of charge carriers in the structure [94,95]. The soliton model can also be applied to the study of dynamics of electrons in macromolecular chains, in the presence of an external electric field [102]. Similarly, in their study of proton dynamics in the molecular chains, Kavitha *et al.* assumed the soliton mechanism of motion of the formed quasi-particle [103]. However, in the case of excitations of a different nature (such as vibronic excitations), it may be necessary to apply an approach in which the semi-classical approximation is not applicable because the vibron and the accompanying deformation form non-adiabatic polaron [90,92,104,105].

Apart from the solitons, certain aspects of the stable migration of electrons along biomolecules, some other mechanisms, can be found in the literature. Such are, for example, models based on continuous super-exchange theory that can provide a good basis for further research on electron migration on shorter segments of DNA [106].

In this section, we study the possibility of forming the stable soliton excitation in molecular systems consisting of two coupled macromolecular chains, like DNA. For this purpose, we analyse the energetic stability of the (possibly formed) soliton, depending on the basic energy parameters of the considered structure. Especially, we consider the influence of the strength of the inter-chain interaction on the stability of the formed soliton and on its distribution between the chains. The consideration is based on the assumption that the excess charge forms an adiabatic large polaron, similarly to the pure 1D macromolecular case.

This section summarizes basic results of research devoted to the possibility of forming an energetically stable soliton excitation in molecular systems consisting of two coupled macromolecular chains, like DNA. A more detailed review of this study (which includes an analysis of the linear stability of the soliton solution) in the case of the DNA-like macromolecule (double-strand structure) is presented in [106], while Chevizovich *et al.* [107,108] present the study of structures consisting of three strands. Here, we have presented an analysis of the energy stability of the (possibly formed) soliton, depending on the basic energy parameters of the considered structure. In particular, we consider the influence of the strength of the inter-chain interaction on the stability of the formed soliton and on its distribution between the chains. The consideration is based on the assumption that the excess charge forms an adiabatic large polaron, similar to the pure 1D macromolecular case.

## 4.1. The model

As a theoretical framework, we consider an extra electron in a structure consisting of two coupled identical macromolecular chains. Each chain is composed by a large number of identical molecular groups, as shown in figure 6. The simplest electron–phonon coupling is described by Holstein Hamiltonian, which is widely used to describe the exciton–phonon systems in the framework of tight

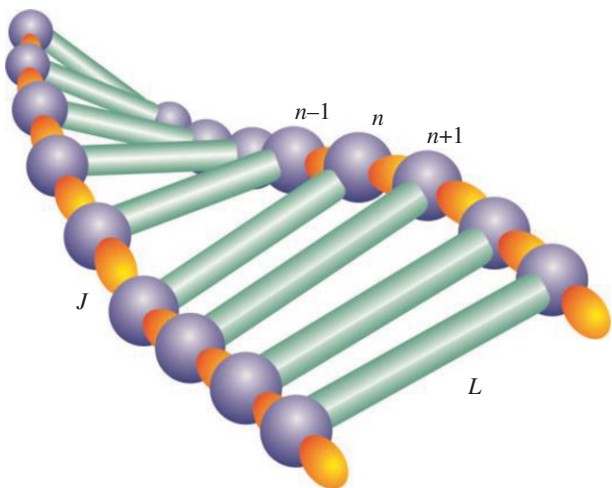

**Figure 6.** Schematic presentation of the part of molecular structure, which consists of two coupled molecular chains.

binding model with on-site interaction among exciton and phonon subsystems [53,88]. For our purpose, we adopted the Holstein model and chose the starting Hamiltonian in the following form:

$$\hat{H} = E_0 \sum_{n,j=1,2} \hat{a}_{n,j}^+ \hat{a}_{n,j} - J \sum_{n,j=1,2} \hat{a}_{n,j}^+ (\hat{a}_{n+1,j} + \hat{a}_{n-1,j}) + L \sum_{n,j=1,2} \hat{a}_{n,j}^+ \hat{a}_{n,3-j} + \frac{1}{\sqrt{N}} \sum_{n,q,j=1,2} F_q e^{iqnR_0} \hat{a}_{n,j}^+ \hat{a}_{n,j} (\hat{b}_{q,j} + \hat{b}_{q,j}^+) + \sum_{q,j=1,2} \hbar \omega_q \hat{b}_{q,j}^+ \hat{b}_{q,j},$$

(4.1)

where $n$ labels the lattice sites, $j = 1$, 2 enumerates molecular chains, $R_0$ is the distance between two adjacent structure elements of the same chain, $E_0$ is the energy of electronic excitation at the $n$th macromolecular site, $L$ is the transversal electron intersite transfer integral (resonant energy of the electron transfer between the nearest molecules at different chains) and $N$ is the number of structure elements in particular chain. Here, we suppose that our electron interacts with acoustic phonon modes [80,82,87,101]. For that reason, exciton–phonon coupling constant $F_q$ has the form $F_q = 2i\chi\sqrt{\hbar/2M\omega_q} \sin qR_0$, where $\omega_q = \omega_0 \sin(qR_0/2)$ is phonon frequency, and $\omega_0 = 2\sqrt{\kappa/M}$. The parameter $\chi$ is the strength of electron–phonon interaction, $\kappa$ is the stiffness of the chain and $M$ is the mass of the molecular group at the site $n$. Here, we suppose that both chains are identical, so electron–phonon coupling constant does not depend on the chain. The $\hat{a}_{n,j}^+$ ($\hat{a}_{n,j}$) are exciton creation (annihilation) operators at the site $n$ and chain $j$, while $\hat{b}_{q,j}^+$ ($\hat{b}_{q,j}$) are phonon creation (annihilation) operators. Finally, $q$ is the phonon wavenumber.

Further examination was done in adiabatic, strong coupling limit, where basic energy parameters of the structure satisfy the following condition [85,87]:

$$2J \gg E_b \gg \hbar \omega_0, \quad E_b = \sum_q \frac{|F_q|^2}{\hbar \omega_q} = \frac{4\chi^2}{M\omega_0^2}.$$

(4.2)

Here, the parameter $E_b$ is the so-called polaron binding energy, and it characterizes the strength of the electron–phonon coupling.

The adiabatic condition ($2J \gg \hbar \omega_0$) provides that the fluctuations in electron and lattice subsystems are uncorrelated and may be neglected. As a consequence, the expectation values of the products of the electron and phonon operators may be factorized and theoretical treatment may be carried out within the semi-classical approximation. Strong coupling condition ($E_b \gg \hbar \omega_0$) provides soliton stability. Under the above circumstances, system dynamics may be described within simple time-dependent extension of the Pekar's variation theory [99–111]. Because of that, the trial state was chosen in the form

$$|\Psi\rangle = \sum_{n,j} \psi_{n,j}(t) \hat{a}_{n,j}^+ |0\rangle_e |\beta(t)\rangle.$$

(4.3)

Here, $\psi_{n,j}$ are exciton amplitudes at the position $n$, and chain $j$, while $|\beta(t)\rangle = \prod_{q,j} |\beta_{q,j}(t)\rangle$ is the phonon state vector which was chosen in the form of multimode coherent phonon state:

$\hat{b}_{q,j}|\beta_{q,j}(t)\rangle = \beta_{q,j}(t)|\beta_{q,j}(t)\rangle$. Due to the conservation of the probability, the exciton amplitudes must satisfy normalization condition $\sum_{n,j}|\psi_{n,j}|^2 = 1$. The $\psi_{n,j}$ and $\beta(t)$ are treated as dynamical variables for which we derive the set of evolution equations minimizing the functional $H = \langle\Psi|\hat{H}|\Psi\rangle$

$$i\hbar\,\dot{\psi}_{n,j} = \Delta\psi_{n,j} - J(\psi_{n+1,j} + \psi_{n-1,j}) + L\psi_{n,3-j} + \frac{1}{\sqrt{N}}\sum_q F_q e^{iqnR_0}\psi_{n,j}(b_{q,j} + b^*_{-q,j}) \qquad (4.4a)$$

and

$$i\hbar\,\dot{\beta}_{q,j} = \hbar\,\omega_q\beta_{q,j} + \frac{1}{\sqrt{N}}\sum_q F_q^* e^{iqnR_0}\psi_{n,j}^*\psi_{n,j}. \qquad (4.4b)$$

In order to study the energetic stability of the soliton formed in the structure, it is necessary to compare its energy with the energy of the free excitation. For this purpose, we first need to analyse the energy spectra of the electron excitation for the case when it does not interact with phonon subsystem (free exciton case). In that case, the last term in equation (4.4a) disappears, and the corresponding solutions of the set of equations can be searched in the form of plane waves $\psi_{n,j}(t) = e^{i(knR_0 - \bar{\omega}(k)t)}A_j$ [106–108]. Here, $k$ is the corresponding wavenumber, $A_j$ is the amplitude that describes the electron distribution over two chains, and $\bar{\omega}(k) = E_{\text{band}}(k)/\hbar$ is normalized energy of the electron excitation in the absence of the electron–phonon interaction (energy of the undressed, bare electron). It is not difficult to show that the energy spectrum of such excitation has the form [112]

$$E_{\text{band}}(k) = E_0 - 2J\cos kR_0 \pm L. \qquad (4.5)$$

In the same time, for the amplitudes, we find $|A_1|^2 = |A_2|^2 = 1/2$ [106]. This means that the electron, due to the inter-chain interaction, becomes equally distributed among the two chains in the absence of the electron–phonon interaction. In the same time, its energy splits into two energy bands: symmetric (+) and antisymmetric (−) one. For that reason, such states are called free (band) exciton states.

Now, we can proceed to seek for soliton solutions, for the system where the excess electron interacts with the phonon subsystem. As was mentioned above, we deal with strong coupling limit, which gives us a hope for the formation of stabile soliton excitation. In addition, in most biological macromolecular structures, this coupling satisfies the condition $2J \gg E_b$. This means that our polaron is spanned over a large number of lattice sites and, consequently, enables us to work in continuum approximation.

In what follows, we derive the set of Hamilton's equations of motion for our dynamical variables. In order to do this, in the first step, we adopt a continuous approximation ($\psi_{n,j}(t) \to \psi_j(x, t)$, $\beta_{q,j}(t) \to \beta_j(x, t)$) and then assume that there exist stable solutions of our electron amplitude functions $\psi_j(x, t) = \psi_j(x - vt)$, where $v$ is the soliton velocity. After solving the obtained equations for phonon amplitudes [101]

$$\beta_{q,j}^{\text{part}}(t) = -\frac{F_{q,j}^*}{\hbar(\omega_q - qv)}\int\frac{dx}{R_0}e^{iqx}|\psi_j(x,t)|^2, \qquad (4.6a)$$

the set of equations for the electron amplitudes takes the following form:

$$i\hbar\dot{\psi}_j + JR_0^2\frac{\partial^2\psi_j}{\partial x^2} + G(v)|\psi_j|^2\psi_j - L\psi_{3-j} = 0, \qquad (4.6b)$$

where [101]

$$G(v) = \frac{4E_b}{1 - v^2/c^2}, \quad c = R_0\omega_0. \qquad (4.7)$$

Equation (4.6b) was obtained by applying an additional transformation $\psi_j(x,t) \to e^{-(i/\hbar)(E_0 - 2J)t}\psi_j(x,t)$. The obtained set equation (4.6b) of two PDEs has the form that is similar to NLSE for single chain case. The difference is only in the last term of the expression, which couples two equations. Therefore, in order to find stable solutions of the system (4.6b), we assume that its solutions do not differ substantially from those of uncoupled ones. More precisely, we assume that the solutions of this system of equations also have a soliton form

$$\psi_j(x,t) = A_j(t)e^{i(Kx - \Omega t)}\phi(\xi), \quad \phi(\xi) = \frac{\phi_0}{\cosh(\sqrt{\alpha}\xi)}, \quad \xi = x - vt, \qquad (4.8)$$

where the complex amplitude $A_j(t)$ is a consequence of the coupling and the function $\phi_0$ is real [106]. The wavenumber, effective mass and carrier wave frequency are, respectively, given by the expressions [106]

$$K = \frac{m^* v}{\hbar}, \quad m^* = \frac{\hbar^2}{2JR_0^2}, \quad \Omega = \frac{\hbar}{JR_0^2}\frac{v^2}{4} - \frac{JR_0^2}{\hbar}\alpha. \tag{4.9}$$

The parameter $\alpha$ is connected to soliton amplitude as $\alpha = (G/2JR_0^2)\phi_0^2$.

The problem of the influence of coupling between the molecular chains on the appearance and stability of the soliton solutions of the system of equations (4.6b) has been reduced to an analysis of the effect of coupling on the amplitudes $A_j(t)$. To do this, it is necessary to express the functional $H$ over amplitudes $A_j(t)$ and to find the corresponding equations of motion, that is [106]

$$H = \varepsilon_0 + a\sum_j A_j^* A_j - \frac{g}{2}\sum_j |A_j|^4 + L\sum_j A_j^* A_{3-j}, \tag{4.10}$$

where

$$\varepsilon_0 = E_0 - 2J, \quad a = \frac{m^* v^2}{2} + \frac{G^2}{48J}, \quad g = \frac{G^2}{12J}\frac{1 - 3v^2/c^2}{1 - v^2/c^2}. \tag{4.11}$$

The corresponding set of equations of motion for the amplitudes is [106]

$$\left.\begin{array}{l} i\hbar\,\dot{A}_1 = aA_1 + LA_2 - g|A_1|^2 A_1 \\ i\hbar\,\dot{A}_2 = aA_2 + LA_1 - g|A_2|^2 A_2 \end{array}\right\}. \tag{4.12}$$

Stationary solutions of the above system of equations are searched in the form $A_j = e^{-i\omega t}B_j$, where $B_j$ are real functions. The character of $\omega$ determines whether the functions $A_j$ represent stable or unstable solutions of equations (4.12). In that sense, the system (4.12) supplemented by a relation originating from the normalization condition attains the following form

$$(\hbar\omega - a)B_1 = LB_2 - gB_1^3, \quad (\hbar\omega - a)B_2 = LB_1 - gB_2^3, \quad B_1^2 + B_2^2 = 1. \tag{4.13}$$

It is not hard to show that the obtained system of equations has two different types of solutions.

### 4.1.1. Asymmetric solution

For $L = -gB_1B_2$, $B_1^2 \neq B_2^2$ we find *asymmetric solutions*

$$B_j = \pm\frac{1}{\sqrt{2}}\sqrt{1 + (-1)^{3-j}\sqrt{1 - 4\gamma^2}}, \quad \hbar\omega = a - g, \tag{4.14}$$

where the parameter $\gamma = |L|/g$ measures the strength of the inter-chain coupling. By the substitution of the obtained solution into the mean value of the Hamiltonian, we find the energy of the asymmetric soliton

$$E_{\text{asym}} = E_0 - 2J + \frac{m^* v^2}{2} + \frac{G^2}{48J} - 2|L|\gamma - \frac{G^2}{48J}\frac{1 - 3v^2/c^2}{1 - v^2/c^2}(1 - 2\gamma^2), \tag{4.15}$$

which, in the non-relativistic limit $v^4/c^4 \ll 1$, becomes

$$E_{\text{asym}} \approx E_0 - 2J + \frac{m_{\text{sol}}v^2}{2} - \frac{G_0^2}{24J}(1 + 4\gamma_0^2), \quad \gamma_0 = \frac{12J|L|}{G_0^2}. \tag{4.16}$$

Here, $m_{\text{sol}} = m^*(1 + G_0^2 R_0^2/6\hbar^2 c^2)$ is the soliton effective mass in the non-relativistic limit. The last term in the expression for $E_{\text{asym}}$ determines the shift of the ground state energy of exciton, and its absolute value determines the large polaron binding energy. As we can remark, the asymmetric solutions exist only for the weak inter-chain coupling regime: $0 < \gamma < 1/2$. These solutions correspond to the soliton predominantly distributed over the single chain.

### 4.1.2. Hybrid (symmetric or antisymmetric) solution

With increasing the strength of the inter-chain interaction, we reach another limit, which is $L \neq gB_1B_2$ and $B_1^2 = B_2^2$. In that case, soliton becomes equally distributed over both macromolecular chains. In the same time, soliton energy spectrum splits into two branches: *symmetric* with both of amplitudes in phase (both

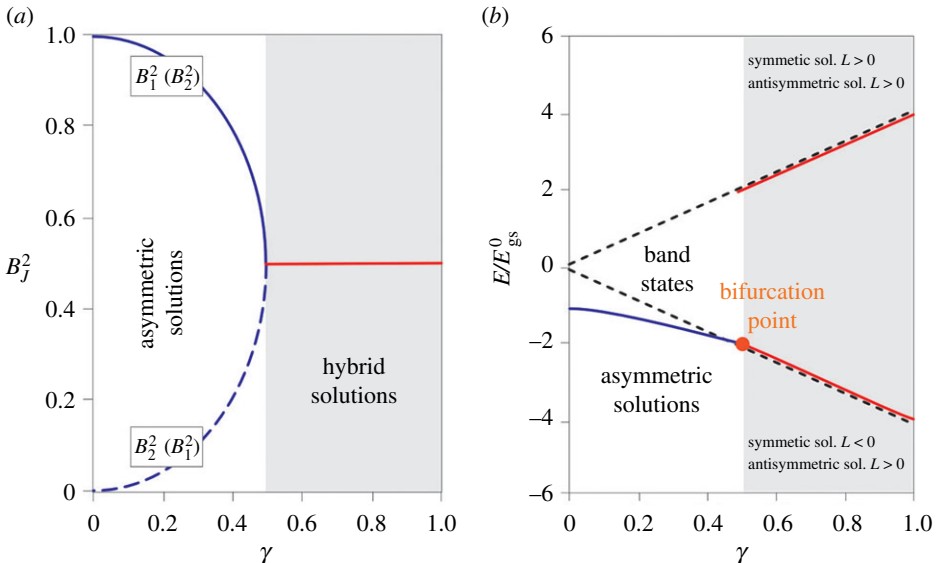

**Figure 7.** (a) The dependence of the square of the soliton amplitudes on the strength of the inter-chain interaction parameter $\gamma$. (b) The dependence of the soliton ground state energy, measured in units $E_{gs}^0 = G_0^2/48J$ (ground state energy in the absence of the inter-chain coupling), on the strength of the inter-chain coupling.

of amplitudes are positive $(++)$ or both of them are negative $(--)$) with $\hbar\omega = a + L - g/2$, and the *antisymmetric* branch that corresponds to solution where soliton amplitudes are out of phase: $(+-)$ or $(-+)$. In that case, we have $\hbar\omega = a - L - g/2$. The energy of 'hybrid soliton' is

$$E_\pm = E_0 - 2J + \frac{m_{sol}v^2}{2} \pm \frac{G^2}{12J}\gamma, \tag{4.17}$$

where $+$ and $-$ correspond to the symmetric and antisymmetric solutions, respectively. Finally, soliton effective mass has the same form as for the asymmetric solutions.

The obtained results are graphically illustrated in figure 7 [106]. On figure 7a, it is presented the dependence of the soliton amplitude on the strength of the inter-chain coupling, for all types of solutions. In the same time, on figure 7b, the energies of both types of the solutions were compared with energy of the electron that does not interact with the phonon subsystem.

From figure 7, we can see that an electron excited on a particular chain due to the presence of an inter-chain interaction can be delocalized to another one. Nevertheless, for the small values of this interaction, it is predominantly localized on the chain where it was excited. Here, we have such solutions that correspond to a soliton dominantly localized on a single chain (asymmetric soliton), which is the true counterpart to the ordinary 1D polaron. But, with the increase in the intensity of this interaction, the probability of its occurrence on the second chain gradually increases, and for $\gamma = 1/2$ soliton becomes equally distributed over both chains. Further increasing the parameter $\gamma$ leads to the emergence of a new type of solutions: hybrid or symmetric (antisymmetric) solitons in which the soliton is equally distributed over both chains, regardless of the strength of the inter-chain interactions.

By comparing the energy of asymmetric soliton solution with the energy of a bare electron, we find that the ground state energy $E_{gs}$ of the asymmetric soliton lies below the bottom of the energy band of the free (band) electron states, which provides the energetic stability of such soliton with respect to the band electron states. On the contrary, the hybrid solitons have energy that is always above the bottom of the band states, so these solutions do not represent stable states of our excitation. As a consequence, the desired criterion for the stable, effectively 1D polaron ($\gamma < 1/2$) may be expressed in the form $E_b^2/(1 - v^2/c^2) > (3/2)LJ$. This condition in the non-relativistic limit attains the form

$$E_b^2 > \frac{3}{2}LJ, \tag{4.18}$$

and is similar to analogous criterion for the large polaron existence within highly anisotropic 3D media that reads [113]

$$E_b^2 > 12LJ. \tag{4.19}$$

The difference in the numerical prefactor in the last equation arises as a consequence of the fact that it is addressed to the highly anisotropic but essentially 3D structure, while the result presented by equation (4.18) concerns the two chain macromolecular system.

Let us mention here that similar results with our prediction previously were reported in the papers devoted to polarons in conducting polymers [114–120]. In the paper of Mizes & Conwell [114], the influence of the inter-chain coupling on the large polaron stability in conducting polymers have been numerically studied within the extended Su–Schrieffer–Heeger (SSH) model. They investigated the polaron properties in a structure consisting of two parallel chains as well as in a cluster composed of a large number of parallel chains. They found that in the case of two chains structure, for weak inter-chain coupling, polaron is practically confined to a single chain. But, with the increasing inter-chain coupling, the depth of the potential well for electron created by the lattice distortion gradually decreases, and electron amplitude becomes equally distributed over both chains.

The presented model sheds additional light on the process of electron migration into the DNA macromolecule and can qualitatively explain the migration of electrons at distances comparable to the dimensions of this macromolecule, a problem that still does not have a satisfactory explanation. The model can be experimentally verified by analysing the phonon spectrum of DNA macromolecules: a self-trapped electron can change the phonon spectrum in a different way than a quasi-free electron.

# 5. Computational challenges and models for DNA dynamics

It is now clear that DNA is a complex molecule with complex interactions. The analytical models for DNA that have been, and are currently, developed (such as the ones described in the previous sections) provide important insight into the behaviour and dynamics of DNA. At the same time, there is an expanding interest and a growing community that is focusing on computational models of DNA. Thanks to the increasing computer power available to research groups, this approach can provide important and complementary views to analytical calculations and experiments. Furthermore, computational methods can provide insight into behaviours of DNA that cannot be computed analytically with current methods, e.g. on topologically complex conformations of DNA that are relevant *in vivo* [121].

There are two major classes of computational models to describe the dynamics of DNA in solution: fully atomistic and coarse-grained. Fully atomistic descriptions reproduce the dynamics of every single atom in a DNA molecule and thus capture the correct set of interactions between nucleotides via, for instance, standard atomistic force-fields such as AMBER [122]. While these models can account for each atom composing nucleotides, they do not account yet for quantum effects such as the ones discussed in §4. In its simplest form, the AMBER force-field accounts for the bonded, angular, dihedral and steric (van der Waals) interactions between atoms by computing the instantaneous potential

$$V(\{r\}) = \sum_{\text{bonds}} k_b(b - b_0)^2 + \sum_{\text{angles}} k_\theta(\theta - \theta_0)^2 + \sum_{\text{dihedrals}} k_\phi[\cos(n\phi + \delta_\phi) + 1] + \sum_{\text{all pairs}(i,j)} \left[ \frac{q_i q_j}{r_{ij}} + \frac{A_{ij}}{r_{ij}^{12}} - \frac{C_{ij}}{r_{ij}^6} \right],$$

(5.1)

where $\{r\}$ denotes the fact that this potential is a function of the position of all the atoms in the system, $b$ is the distance between bonded atoms, $\theta$ is the angle formed by two vectors joining two consecutive pairs of atoms, $\phi$ is the angle between two planes defined by two triplets of atoms and $r_{ij}$ is the distance between non-bonded atoms. Equation (5.1) is then used to compute the force experienced by each atom in the system and evolve their equations of motion with implicit, explicit or absence of solvent particles. The downside of full-atom simulations is that they currently struggle to encompass the time and length scales necessary to provide insight into the dynamics of DNA in realistic conditions, as it can be formed by millions of nucleotides (about 100 million in the case of whole human chromosomes). Nonetheless, it should be noted that as computer power increases, some large-scale full-atom simulations of DNA approximately hundreds of base pairs long are becoming available and can be comparable to experimental set-ups [123,124]. These studies are mostly focused on the role of supercoiling (and very recently, protein binding [125]) in the structural and 3D conformation of DNA loops and are providing important insight into processes which are difficult to capture experimentally, such as base flipping. Opening of the DNA double helix, or DNA denaturation, can also be observed in these simulations [126], but the short size of the DNA loops is a limiting factor as finite-size effects can be important.

In order to reach DNA sizes and time-scales that are realistic *in vivo*, it is common to employ coarse-grained DNA models. This approach groups tens of atoms into rigid bodies (usually spherical, for computational efficiency) which then interact with each other through effective potentials. The downside is that the parameters are no longer based on inter-atom interactions and need to be carefully parametrized in order to reproduce realistic values of measurable observables such as DNA persistence length. The first generation of such models were based on a semiflexible polymer with large persistence length (compared with its thickness $a \approx 2.5$ nm, DNA has a persistence length $l_p = 20a$) and with torsional rigidity [127,128] which could give rise to so-called supercoiled conformations, such as the ones adopted by a writhing telephone cord. Importantly, these early models neglected the double-stranded nature of DNA and they could not describe base-pair opening, DNA melting or phenomena such as the ones predicted by Peyrard–Bishop-type models (see previous sections). To fill in this gap, more sophisticated computer models needed to be developed.

Recently, two such models have been proposed. The first, commonly known as 'oxDNA' [129] (because developed by a group in Oxford), is a single-nucleotide resolution model which coarse-grains all the atoms within a nucleotide into one spherical bead representing the sugar-phosphate part and one pair of beads representing the base and its orientation. Note that the level of this coarse-graining is very similar to the one typically adopted in analytical models such as the PB modes described above. In this model, the interactions between the nucleotides belonging to the same strand are different from the interactions between nucleotides in different strands, thereby allowing for single-strand connectivity and stacking. Note that in these numerical models, it possible to fully account for stacking, while it had to be neglected in analytical models (see §3). The potential between nucleotides can be written as

$$V = \sum_{\text{nn single strand}} (V_{\text{backbone}} + V_{\text{stack}} + V'_{\text{steric}}) + \sum_{\text{all other}} (V_{\text{HB}} + V_{\text{cross-stack}} + V_{\text{steric}}), \tag{5.2}$$

where $V_{\text{backbone}}$ is a finitely extensible nonlinear elastic (FENE) spring that keeps consecutive nucleotides together at a distance $d_{bb}$ representing covalent bonds, $V_{\text{stack}}$ is a Morse potential which maintains bases together at distance $d_{bp}$ and aligned, and nn stands for nearest neighbours. Importantly, setting $d_{bp} < d_{bb}$ naturally drives the double strand to arrange into a helix; the right-handed chirality is set manually by reducing the strength of the interaction for increasing left-hand twist. $V_{\text{steric}}$ and $V'_{\text{steric}}$ account for excluded volume interactions and are different for neighbours within the single strand and neighbours in different strands because of the presence of an FENE potential. $V_{\text{HB}}$ represents the hydrogen bonding between bases and can be written as a Morse potential, in the same form as the one employed in the PB model in equation (3.5), which tends to maintain the nucleotides at a given distance from each other. In this computational model, $V_{\text{HB}}$ is coupled to a smoothing function and a modulation that restrict this interaction to bases that are anti-aligned, in order to reproduce the $3' \to 5'$ directionality encoded in the DNA. Finally, $V_{\text{cross-stack}}$ is an interaction between a base and the neighbour base in the opposite strand and it ensures stabilization of the structure. The full form of the potentials, smoothing functions and modulation terms are described in [129,130] for its implementation in a widely employed simulations engine (see also below). This model required extensive characterization to set its approximately 60 parameters in order to reproduce known features of the DNA double helix, but after this mapping was accomplished and extensively tested [131–134] it has been widely adopted by the scientific community to study the behaviour of DNA. Particularly notable is the ability of oxDNA to describe DNA hybridization and hairpins [3,134–136], which is useful for applications to *in vitro* technology such as DNA origami [137].

Of special interest to this review are recent works that employed oxDNA to study the interplay of twisting and bending on the conformation assumed by a DNA molecule [138–141]. In more detail, it was discovered that the structural anisotropy of DNA, i.e. the presence of the so-called major and minor grooves in between the phosphate atoms on opposite nucleotides in a base-pair (figure 5), endows the chain with an effective coupling between its bending and twisting modes. In the language of the previous sections, for instance, in the helicoidal PB model, the twisting degree of freedom is weakly coupled to the bending of the helix. Curiously, the first generation of the oxDNA model did not account for this structural anisotropy and no coupling could be observed; this was instead successfully recovered in oxDNA 2.0, which correctly describes the structural anisotropy of the helix [137,142]. The physical implications of this coupling are profound and mechanically akin to those described in §3. For instance, if a protein such as RNAP (figure 5) bends the DNA locally, this mechanical deformation propagates in the form of mechanical twist waves down the chain [138]. These waves are reminiscent of soliton solutions in the PB-like models described above and have

important implications for the stability of nucleosomes, the complex of DNA and histone proteins, which makes the chromatin in eukaryotic cells [138]. It should be noted that while this coupling can be effectively accounted for through purely elastic (quadratic) terms in the Hamiltonian [8,137], the numerical studies were performed with a computational model that accounted for highly nonlinear interactions such as stacking and cross-stacking via Morse potentials.

While oxDNA has been useful for reaching a deeper understanding of, for instance, DNA structure and hybridization kinetics [3], it has rarely been applied to tackling phenomena that are relevant *in vivo* such as DNA folding (for instance, around proteins) or melting. These ubiquitous processes have recently been tackled using another single base-pair resolution model for DNA [143]. This model has been developed independently from oxDNA and has the advantage of having fewer parameters to be set manually: it represents the sugar-phosphate backbone as a series of spheres with 'patches' which represent the bases. The dynamics of the complex sphere-patch is numerically evolved as a rigid body and spheres interact only sterically and are bonded with their consecutive neighbours along the same DNA strand. The patches of bonded bases are constrained with a truncated harmonic potential

$$V(r) = \frac{K_2}{2(r_0 - r_c)^2}((r - r_0)^2 - (r_c - r_0)^2),$$ (5.3)

if $r \leq r_c$ and 0 otherwise. The minimum of this potential is at $r_0$, but the bond breaks if $r > r_c$. This allows this model to capture faithfully DNA denaturation. The fact that it is implemented in LAMMPS (Large-scale Atomic/Molecular Massively Parallel Simulator, http://lammps.sandia.gov), a highly efficient and scalable engine for large-scale parallel molecular dynamics simulations [144] renders this model particularly apt to simulate large DNA molecules. Note that recently also oxDNA has been made available to be used in LAMMPS [130]. More specifically, it has been used to study the competition between DNA opening (such as the one described in the previous sections) and DNA supercoiling due to the invariance of the linking number between the two DNA strands within closed plasmids [145]. In this case, the topology of DNA, a property that is invariant under smooth deformations of its contour [146], is key to establish this competition. For instance, while it is broadly known that the melting transition of DNA is cooperative, or first-order-like [147], circular DNA displays a much smoother melting transition [148]. The simulations also reveal that there is a broad coexistence region (in thermal equilibrium), whereby a DNA molecule is not either denatured or closed, but it can actually display the simultaneous coexistence of denatured and closed regions along the same molecule [148]. In other words, introducing a topological constraint in the problem (which has instead been neglected in the previous sections) profoundly affects this thermal phenomenon and it would be thus interesting to explore further whether it may affect mechanical ones such as the ones discussed in the previous sections. In general, understanding qualitative difference in the mechanics of DNA due to its topological nature (linear or open versus circular or closed) would require a theoretical description that can account for the invariance of the total linking number (Lk), connected to the twist (Tw) and writhe (Wr) via the famous White–Fuller–Calugareanu theorem

$$Lk = Tw + Wr$$ (5.4)

stating that the linking between the two DNA strands must sum up to twist Tw, i.e. the signed crossing of two single-strands, and writhe Wr, i.e. the signed crossing of DNA centreline [121]. It should be highlighted that the phenomenon of DNA melting is thermal in nature and should not be confused with the mechanical opening of base pairs described in §3 leading to soliton solutions or breathers. On the other hand, the two effects, mechanical and thermal, may act concomitantly, especially in living cells where we have seen that RNAP contributes to mechanically open the DNA duplex. Furthermore, the recent coupling between twisting and bending modes revealed using the oxDNA model, or others suggested by PB-like models, may play an important role in this and other thermal phenomena. Finally, to the best of the authors' knowledge, Peyrard–Bishop-like models have not yet been extended to account for topological constraints and, therefore, this may be an interesting theoretical extension for the future.

# 6. Conclusion and discussions

We reviewed some of the existing theoretical models to describe nonlinear DNA dynamics, and the HPB model was elucidated in some more details. Within this model, the semi-discrete approximation as a mathematical tool was used and explained. It was shown that it yields to the so-called breather

solutions, which are local opening (or denaturation) bubbles moving along the DNA backbone. It is important to point out that continuum approximation brings about kink solitons moving along the DNA chain, as was shown recently [149].

Additionally, we presented a soliton model that, in principle, can explain an important question concerning the stabile long-distance charge transport in DNA molecule. We discussed the limitations of its applicability, due to the presence of the inter-chain interaction that may break the stability of the eventually formed soliton.

Alongside theoretical advances, we discussed the computational advances that the communities working on DNA dynamics and efficient large-scale computing have been making. They developed a series of models, ranging from fully atomistic descriptions to more coarse-grained ones, which can encompass orders of magnitude in DNA lengths and time-scales investigated. These models are currently extensively being used to understand complex processes such as DNA hybridization and denaturation, where local opening of DNA base pairs is key. In the future, we envisage that some of these models may be employed to test the 'breathers' solutions discussed in this review at the theoretical level on a more realistic situation, such as that of a DNA molecule diffusing in 3D space and perhaps subject to topological constraints such as knots or closed into a loop.

We finally highlight that any model is ultimately as good as its ability to make predictions that can be experimentally tested. At present, many different experimental methods for studying the structure and dynamics of DNA have been developed. The most important are X-ray diffraction analysis, neutron scattering, infrared (IR) spectroscopy, hydrogen–deuterium(–tritium) exchange, resonant microwave absorption and nuclear magnetic resonance (NMR) [150,151]. In spite of this, fluorescence correlation spectroscopy of molecular beacons is the only technique that allows studying the kinetics of the denaturation bubble in DNA [152–154]. Unfortunately, these experiments can neither prove nor disprove the results of the theoretical investigations mentioned above. For example, they cannot detect nonlinear waves, check their nature, measure their speed and width, and so on. However, there is hope that micro-manipulating experiments on a single DNA molecule could provide a strategy to check the theoretical predictions explained in this review. Namely, a new era of experimental research has started from the end of the last century. This means that mechanical manipulations on a single molecule, that are stretching, twisting and untwisting, are now possible [155–165]. While the thermodynamics of these manipulations have been extensively studied [166–168], there are still ample opportunities for kinetic studies aimed at detecting emergent DNA behaviours as well as DNA–protein interactions, which could be modelled within the frameworks reviewed in this paper.

Data accessibility. This article has no additional data.

Authors' contributions. D.C. and D.M. wrote §§4 and 5, respectively. The remaining three authors were engaged in writing the rest of the manuscript. S.Z. wrote §3.

Competing interests. We declare we have no competing interests.

Funding. D.C., D.M. and S.Z. would like to acknowledge networking support by the COST Action CA17139. D.M. is supported by the Leverhulme Trust through an Early Career Fellowship (ECF-2019-088).

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
