## [Reviewer comments · Royal Society Open Science]

Review History

RSOS-200774.R0 (Original submission)

Review form: Reviewer 1

Is the manuscript scientifically sound in its present form?

Yes

Are the interpretations and conclusions justified by the results?

Yes

Is the language acceptable?

Yes

Do you have any ethical concerns with this paper?

No

Have you any concerns about statistical analyses in this paper?

No

Recommendation?

Major revision is needed (please make suggestions in comments)

Comments to the Author(s)

The review reports on different aspects of DNA dynamics and stability, with special focus on theoretical frameworks that reproduce the emergence of breathing modes and explain the charge and energy transport along the chain. There is also a section of the numerical modelling of DNA. Unfortunately, the overall feeling is that the manuscript reads as a collection of different parts rather than a single review revolving around a specific topic.

Sections II and III are the most coherent with the title, reporting a review of some of the theoretical models used to investigate the stability of double-stranded DNA with classical theories (although it should be noted that these sections reference 13 papers authored by the corresponding author, which makes this part of the review appear a bit too self-centred).

By contrast, in Section IV the authors discuss phenomena which have a quantum origin.

However, differently from the previous sections, most of the work reported here seems original, or at least it is not clear whether the theory and results reported herein are new or reproduced from some other work (e.g. Figure 5). This in stark contrast with section III, which reports work already presented elsewhere.

Finally, Section V reviews some of the computational models used to investigate the structure and dynamics of DNA at different coarse-graining levels. However, the connection with the previous sections is tenuous at best. In fact, there is no connection with Section IV at all, whereas the discussion about the appearance of bubbles in simulations is only sketched.

All in all, I cannot recommend publication in its current form, as the manuscript does not feel as a single unit, and the inter-connections between the single parts are not strong enough to warrant their inclusion in a single review. The authors should make the effort to uniform the language and the style and to add more overlap between the different sections.

Here is a list of minor things:

- The typesettings of some equations are a bit off (at least on my pdf reader). For instance, some vector signs and other symbols are not displayed correctly (e.g. eqq 2.4, 4.4a and b, 4.6b, eq 4.9).
- Stacking is at least as important as (if not more important than) hydrogen bonding in dictating the stability of the duplex (see e.g. <https://academic.oup.com/nar/article/34/2/564/2401647>)
- I think a figure describing the theories introduced in Sections II and III would greatly help the reader grasping the nature of the models presented.
- I guess the sums in Section III run over all the nucleotides, but this is not specified in the text and in the equations.
- oxDNA has been used to investigate DNA melting (see e.g. <https://doi.org/10.1093/nar/gkt687>).
- On pag 23 I believe that Ref [130] should be Ref [129].

Review form: Reviewer 2 (Victor Lakhno)

Is the manuscript scientifically sound in its present form?

Yes

Are the interpretations and conclusions justified by the results?

Yes

Is the language acceptable?

Yes

Do you have any ethical concerns with this paper?

No

Have you any concerns about statistical analyses in this paper?

No

Recommendation?

Accept with minor revision (please list in comments)

Comments to the Author(s)

The review considers nonlinear dynamics of DNA and is actual. In the first part of review the authors consider several models of DNA dynamics without electron. I recommend authors to read and refer the detailed review on this subject [1].

In the second part the authors consider the long-range polaron transport in DNA. This problem was also considered in paper [2] which need to be cited.

With these improvements I recommend the paper for publication in Royal Society Open Science.

[1] Shigaev A.S. et al. "Theoretical and Experimental Investigations of DNA Open States". *Math. Biol. Bioinf.* 2018;13(Suppl.):t162-t267 <https://doi.org/10.17537/2018.13.t162>

[2] Lakhno, V. "Soliton-like Solutions and Electron Transfer in DNA". *Journal of Biological Physics* 26, 133-147 (2000). <https://doi.org/10.1023/A:1005275211233>

Decision letter (RSOS-200774.R0)

Dear Dr Zdravkovic

The Editors assigned to your paper RSOS-200774 "A review of recent studies on the nonlinear dynamics of DNA" have now received comments from reviewers and would like you to revise the paper in accordance with the reviewer comments and any comments from the Editors. Please note this decision does not guarantee eventual acceptance.

Please submit your revised manuscript and required files (see below) no later than 21 days from today's (ie 10-Sep-2020) date. Note: the ScholarOne system will 'lock' if submission of the revision is attempted 21 or more days after the deadline. If you do not think you will be able to meet this deadline please contact the editorial office immediately.

on behalf of the Associate Editor, and Professor Pietro Cicuta (Subject Editor)
openscience@royalsociety.org

Associate Editor Comments to Author:

Thank you for your patience while your manuscript was reviewed. Two referees have commented on your paper - the first offers substantive feedback that we'd like you to address in a revision. The second offers a pair of references you may wish to consider including - please note, however, that the editors observe both references are from one author and it is not clear to the editors how critical these reference suggestions are. Please only include these references if you feel it adds value to your work and they can be critically considered in the review - otherwise, please consider disregarding their inclusion as a potential example of 'citation stacking'. This practice is not to be encouraged.

Reviewer comments to Author:

Reviewer: 1
Comments to the Author(s)

The review reports on different aspects of DNA dynamics and stability, with special focus on theoretical frameworks that reproduce the emergence of breathing modes and explain the charge and energy transport along the chain. There is also a section of the numerical modelling of DNA.

Unfortunately, the overall feeling is that the manuscript reads as a collection of different parts rather than a single review revolving around a specific topic.

Sections II and III are the most coherent with the title, reporting a review of some of the theoretical models used to investigate the stability of double-stranded DNA with classical theories (although it should be noted that these sections reference 13 papers authored by the corresponding author, which makes this part of the review appear a bit too self-centred).

By contrast, in Section IV the authors discuss phenomena which have a quantum origin. However, differently from the previous sections, most of the work reported here seems original, or at least it is not clear whether the theory and results reported herein are new or reproduced from some other work (e.g. Figure 5). This in stark contrast with section III, which reports work already presented elsewhere.

Finally, Section V reviews some of the computational models used to investigate the structure and dynamics of DNA at different coarse-graining levels. However, the connection with the previous sections is tenuous at best. In fact, there is no connection with Section IV at all, whereas the discussion about the appearance of bubbles in simulations is only sketched.

All in all, I cannot recommend publication in its current form, as the manuscript does not feel as a single unit, and the inter-connections between the single parts are not strong enough to warrant their inclusion in a single review. The authors should make the effort to uniform the language and the style and to add more overlap between the different sections.

Here is a list of minor things:

- The typesettings of some equations are a bit off (at least on my pdf reader). For instance, some vector signs and other symbols are not displayed correctly (e.g. eqq 2.4, 4.4a and b, 4.6b, eq 4.9).
- Stacking is at least as important as (if not more important than) hydrogen bonding in dictating the stability of the duplex (see e.g. <https://academic.oup.com/nar/article/34/2/564/2401647>)
- I think a figure describing the theories introduced in Sections II and III would greatly help the reader grasping the nature of the models presented.
- I guess the sums in Section III run over all the nucleotides, but this is not specified in the text and in the equations.
- oxDNA has been used to investigate DNA melting (see e.g. <https://doi.org/10.1093/nar/gkt687>).
- On page 23 I believe that Ref [130] should be Ref [129].

Reviewer: 2

Comments to the Author(s)

The review considers nonlinear dynamics of DNA and is actual. In the first part of review the authors consider several models of DNA dynamics without electron. I recommend authors to read and refer the detailed review on this subject [1].

In the second part the authors consider the long-range polaron transport in DNA. This problem was also considered in paper [2] which need to be cited.

With these improvements I recommend the paper for publication in Royal Society Open Science.

[1] Shigaev A.S. et al. "Theoretical and Experimental Investigations of DNA Open States". *Math. Biol. Bioinf.* 2018;13(Suppl.):t162-t267 <https://doi.org/10.17537/2018.13.t162>

[2] Lakhno, V. "Soliton-like Solutions and Electron Transfer in DNA". *Journal of Biological Physics* 26, 133-147 (2000). <https://doi.org/10.1023/A:1005275211233>

===PREPARING YOUR MANUSCRIPT===

- one version identifying all the changes that have been made (for instance, in coloured highlight, in bold text, or tracked changes);
- a 'clean' version of the new manuscript that incorporates the changes made, but does not highlight them. This version will be used for typesetting if your manuscript is accepted.

===PREPARING YOUR REVISION IN SCHOLARONE===

-- Ensure that your data access statement meets the requirements at <https://royalsociety.org/journals/authors/author-guidelines/#data>. You should ensure that you cite the dataset in your reference list. If you have deposited data etc in the Dryad repository, please include both the 'For publication' link and 'For review' link at this stage.

Author's Response to Decision Letter for (RSOS-200774.R0)

See Appendices A & B.

RSOS-200774.R1 (Revision)

Review form: Reviewer 1

Is the manuscript scientifically sound in its present form?

Yes

Are the interpretations and conclusions justified by the results?

Yes

Is the language acceptable?

Yes

Do you have any ethical concerns with this paper?

No

Have you any concerns about statistical analyses in this paper?

No

Recommendation?

Accept as is

Comments to the Author(s)

The revised version of the manuscript has been significantly improved by the authors, and while the review is still very broad in context, the authors explicitly state in the text why this is the case. In addition, they have also strengthened the connection between the sections and changed the title. I am happy to recommend publication.

Decision letter (RSOS-200774.R1)

Dear Dr Zdravkovic,

It is a pleasure to accept your manuscript entitled "A Review on Nonlinear DNA Physics" in its current form for publication in Royal Society Open Science. The comments of the reviewer(s) who reviewed your manuscript are included at the foot of this letter.

on behalf of the Associate Editor and Professor Pietro Cicuta (Subject Editor)
openscience@royalsociety.org

Reviewer comments to Author:

Reviewer: 1
Comments to the Author(s)

The revised version of the manuscript has been significantly improved by the authors, and while the review is still very broad in context, the authors explicitly state in the text why this is the case.

In addition, they have also strengthened the connection between the sections and changed the title. I am happy to recommend publication.

Appendix A

Editor – Answer

For Editor's text small and bold letters are used.

Associate Editor Comments to Author:

Two referees have commented on your paper - the first offers substantive feedback that we'd like you to address in a revision. The second offers a pair of references you may wish to consider including - please note, however, that the editors observe both references are from one author and it is not clear to the editors how critical these reference suggestions are. Please only include these references if you feel it adds value to your work and they can be critically considered in the review - otherwise, please consider disregarding their inclusion as a potential example of 'citation stacking'. This practice is not to be encouraged.

Thank you for encouraging us not to strictly follow the reviewer's requirements. Authors are in dilemma sometimes either to follow the requirements with which they do not agree or their personal opinions. We belong to the second group of the authors but it is certainly relaxing to obtain the comment like the Editor's above.

While writing our manuscript, we tried to escape overlapping with the references suggested by Reviewer 2, especially with Ref. [1], which is larger review paper than our one. However, this is not reason not to mention them and, we think, we should have relied on them a little bit. Hence, we include these references in our manuscript believing that they add value to our work.

We thank the Editor for his work on our manuscript. We believe that the revised version is good enough to be published in RSOS.

Sincerely,
The authors

Appendix B

Reviewer 1 – Answer

For Reviewer's text small and bold letters are used. **New text in our revised manuscript is blue.**

Unfortunately, the overall feeling is that the manuscript reads as a collection of different parts rather than a single review revolving around a specific topic.

We thank the referee for this comment. Our intention is to cover different problems within the broad field of DNA non-linear dynamics, to give a wide overview of the field. In particular, Sections 2 and 3 are free from moving charges and, consequently, free from quantum mechanics. In Section 4 free electrons are present, requiring quantum mechanical approach. While Section 4 may seem uncorrelated from the rest, it should be interpreted as a further layer of complexity that is added on top of the ones described in Sections 2 and 3. In particular, the model we discuss couples the motion of free electrons to the mechanical modes of the DNA structure thus creating a bridge between Section 4 and the others. We now highlight this connection in more detail in the revised text.

In Section 5 we review the state-of-the-art of numerical modelling of DNA mechanics and discuss in more detail recent numerical works that uncovered a coupling between twisting and bending modes in DNA. For instance, bending of the DNA generates twist waves that are reminiscent of the soliton solutions in the PB model. We also discuss that simulations can account for topological constraints, which are usually neglected in analytical models and may profoundly affect non-linear DNA dynamics. Accounting for topological constraints may be an interesting way forward for analytical models.

More generally, the composition of our review aims to target readers that are looking to familiarize themselves with the broad literature of DNA non-linear dynamics. Indeed, we feel that our review will satisfy a reader looking for a comprehensive review that covers not only the most popular analytical methods to treat DNA non-linearities but also provides a broader context and information on quantum and computational methods. In the revised paper we have focused on strengthening the connections between sections so that a reader will feel that they are parts of a bigger picture. We finally thank the referee for pushing us to improve our work in this direction.

Sections II and III are the most coherent with the title, reporting a review of some of the theoretical models used to investigate the stability of double-stranded DNA with classical theories (although it should be noted that these sections reference 13 papers authored by the

corresponding author, which makes this part of the review appear a bit too self-centred).

The title has been changed. Regarding the references, the name Gaeta appears 13 times among references, Peyrard 12, Yakushevich 7, Dauxois 6, Daniel 6 times, etc. We want to point out that we give a short overview of a couple of models and explain one in detail. The one we choose to discuss in detail requires a few more references and is the one in which the authors of this work are most expert on.

By contrast, in Section IV the authors discuss phenomena which have a quantum origin. However, differently from the previous sections, most of the work reported here seems original, or at least it is not clear whether the theory and results reported herein are new or reproduced from some other work (e.g. Figure 5). This in stark contrast with section III, which reports work already presented elsewhere.

As we now stress in the revised paper, Section 4 is meant to give an overview of quantum methods that can be used to tackle the problem of charge transport on DNA or other macromolecules. In particular, the model of “self-trapping” electrons that we discuss offers a connection between charge movement and structural DNA dynamics (treated in the other Sections). We should point out that all results in Section 4 were reported in the literature. This is now indicated in the revised manuscript.

Finally, Section V reviews some of the computational models used to investigate the structure and dynamics of DNA at different coarse-graining levels. However, the connection with the previous sections is tenuous at best. In fact, there is no connection with Section IV at all, whereas the discussion about the appearance of bubbles in simulations is only sketched.

To the best of our knowledge, there is no computational model to simulate quantum effects on DNA or other polymers (arguably it will be developed alongside quantum computing in the near future). At the same time, summarizing the numerical state-of-the-art to simulate DNA at “all atom” and “coarse-grained” level is important in this review, as it provides the unfamiliar reader with basic references and concepts that s/he can further pursue if interested within the broad field of DNA mechanics. In particular, we now have made a direct connection between this section and the others by discussing in detail the recent finding of twist waves in oxDNA models. These waves are reminiscent of the soliton solutions found in PB-like models in Secs. 2 and 3 and would be interesting to explore this further in the future.

In this Section we also discuss how numerical models can offer additional insight with respect to analytical models, for instance by fully accounting for topological constraints. We now have also followed the suggestion of the referee and have

expanded the discussion on simulated DNA melting. We thank the referee for this useful remark.

All in all, I cannot recommend publication in its current form, as the manuscript does not feel as a single unit, and the inter-connections between the single parts are not strong enough to warrant their inclusion in a single review. The authors should make the effort to uniform the language and the style and to add more overlap between the different sections.

In the revised paper we have made an effort to highlight the connections between Sections and now clearly explain our aim and target audience in the introduction. We feel that with these edits, the manuscript should warrant publication.

Here is a list of minor things:

- The typesettings of some equations are a bit off (at least on my pdf reader). For instance, some vector signs and other symbols are not displayed correctly (e.g. eqq 2.4, 4.4a and b, 4.6b, eq 4.9).

This should be OK after fixing at editorial stage.

- Stacking is at least as important as (if not more important than) hydrogen bonding in dictating the stability of the duplex (see e.g. <https://academic.oup.com/nar/article/34/2/564/2401647>)

Stacking is certainly very important. It is difficult to compare it with hydrogen bonding because the breathing of nucleotide pairs is referred to the hydrogen bond. In other words, Fig.4 shows the nucleotide pair stretching, which is, of course, in the direction of the hydrogen bond.

Our model, that is Hamiltonian (3.6), does not include any term describing the stacking. The stacking interaction is between the neighbouring nucleotides belonging to the same strand. The same holds for the covalent bond. The later one is much stronger and one can say that the stacking is neglected. By the way, there is not any model comprising all interactions. To the best of our knowledge, there is not any model describing the stacking that would fit to our review. Even though the stacking is weaker than the covalent bond, we do not think that the neglecting it is completely OK, because the natures of these interactions are different. We believe that the HPB model, or any other, should be improved including one more term in Eq. (3.6) or equivalent. As the stacking is not as strong as covalent interaction, this term should be an enharmonic one. We can speculate now that one possibility could be Morse

potential that includes $u_n - u_{n-1}$ and $v_n - v_{n-1}$ instead of $u_n - v_n$, existing in Eq.(3.6). Of course, the question will be, as usual, if this case is analytically solvable. Therefore, the problem with the stacking is very interesting but should be postponed for the future research. This is indicated in the revised manuscript shortly.

- I think a figure describing the theories introduced in Sections II and III would greatly help the reader grasping the nature of the models presented.

We agree. Two figures have been introduced in Sections 2 and 3.

- I guess the sums in Section III run over all the nucleotides, but this is not specified in the text and in the equations.

Correct. This is specified in the revised manuscript.

- oxDNA has been used to investigate DNA melting (see e.g. <https://doi.org/10.1093/nar/gkt687>).

The reference mentioned by the referee is on DNA hybridization and zippering, i.e. the joining of two single DNA strands. On the other hand, DNA melting is the opposite, i.e. the opening of the DNA duplex. Nevertheless, we have now expanded and searched for more references on numerical studies of DNA melting with coarse-grained models.

- On page 23 I believe that Ref [130] should be Ref [129].

Thank you. This is corrected.

Finally, we want to thank the unknown reviewer for his/her useful suggestions.

Sincerely,
The authors